# Cathepsin S Evokes PAR_2_-Dependent Pain in Oral Squamous Cell Carcinoma Patients and Preclinical Mouse Models

**DOI:** 10.3390/cancers13184697

**Published:** 2021-09-19

**Authors:** Nguyen Huu Tu, Kenji Inoue, Elyssa Chen, Bethany M. Anderson, Caroline M. Sawicki, Nicole N. Scheff, Hung D. Tran, Dong H. Kim, Robel G. Alemu, Lei Yang, John C. Dolan, Cheng Z. Liu, Malvin N. Janal, Rocco Latorre, Dane D. Jensen, Nigel W. Bunnett, Laura E. Edgington-Mitchell, Brian L. Schmidt

**Affiliations:** 1Bluestone Center for Clinical Research, Department of Oral and Maxillofacial Surgery, New York University (NYU) College of Dentistry, New York, NY 10010, USA; n.h.tu@nyu.edu (N.H.T.); ki630@nyu.edu (K.I.); ec128@nyu.edu (E.C.); cs6135@nyu.edu (C.M.S.); nns18@pitt.edu (N.N.S.); hdt222@nyu.edu (H.D.T.); dhk422@nyu.edu (D.H.K.); rga283@nyu.edu (R.G.A.); ly2176@nyu.edu (L.Y.); jcd10@nyu.edu (J.C.D.); ddj3@nyu.edu (D.D.J.); 2Department of Biochemistry and Pharmacology, Bio21 Institute, University of Melbourne, Parkville, VIC 3052, Australia; bethany@student.unimelb.edu.au; 3Hillman Cancer Research Center, Department of Neurobiology, University of Pittsburgh, Pittsburgh, PA 15232, USA; 4Pathology Department, New York University (NYU) Langone Health, New York, NY 10016, USA; cheng.liu@nyulangone.org; 5Department of Epidemiology and Health Promotion, New York University (NYU) College of Dentistry, New York, NY 10010, USA; mj62@nyu.edu; 6Department of Molecular Pathobiology, New York University (NYU) College of Dentistry, New York, NY 10010, USA; rl3423@nyu.edu (R.L.); nwb2@nyu.edu (N.W.B.); 7Department of Neuroscience and Physiology, Neuroscience Institute, New York University (NYU) Langone Health, New York, NY 10016, USA; 8Drug Discovery Biology, Monash Institute of Pharmaceutical Sciences, Monash University, Parkville, VIC 3052, Australia

**Keywords:** oral cancer, pain, cathepsin S, protease-activated receptor-2, PAR_2_, cancer pain, oral squamous cell carcinoma

## Abstract

**Simple Summary:**

Oral cancer is often deadly and severely painful. Because this form of cancer pain cannot be adequately treated with current medications including opiates, new treatment approaches are needed. Cathepsin S, a lysosomal cysteine protease may play a role in oral cancer pain through a protease-activated receptor-2 (PAR_2_)-dependent mechanism. We undertook a series of experiments to define the role of cathepsin S in oral cancer pain. We compared cathepsin S activity in human oral cancer tumors versus patient-matched normal tissue; a human oral cancer cell line versus a benign dysplastic oral keratinocyte cell line; and in an orthotopic xenograft tongue cancer mouse model versus normal controls in mice. We localized cathepsin S in macrophages and carcinoma cells in human oral cancers. The injection of cathepsin S caused nociception in a mouse model while the injection of oral cancer cells in which the gene for cathepsin S is deleted generated less nociception. Our findings will lay the foundations for clinical trials of cathepsin S inhibitors for treating oral cancer pain.

**Abstract:**

Oral squamous cell carcinoma (SCC) pain is more prevalent and severe than pain generated by any other form of cancer. We previously showed that protease-activated receptor-2 (PAR_2_) contributes to oral SCC pain. Cathepsin S is a lysosomal cysteine protease released during injury and disease that can activate PAR_2_. We report here a role for cathepsin S in PAR_2_-dependent cancer pain. We report that cathepsin S was more active in human oral SCC than matched normal tissue, and in an orthotopic xenograft tongue cancer model than normal tongue. The multiplex immunolocalization of cathepsin S in human oral cancers suggests that carcinoma and macrophages generate cathepsin S in the oral cancer microenvironment. After cheek or paw injection, cathepsin S evoked nociception in wild-type mice but not in mice lacking PAR_2_ in Na_v_1.8-positive neurons (Par_2_Na_v_1.8), nor in mice treated with LY3000328 or an endogenous cathepsin S inhibitor (cystatin C). The human oral SCC cell line (HSC-3) with homozygous deletion of the gene for cathepsin S (*CTSS*) with CRISPR/Cas9 provoked significantly less mechanical allodynia and thermal hyperalgesia, as did those treated with LY3000328, compared to the control cancer mice. Our results indicate that cathepsin S is activated in oral SCC, and that cathepsin S contributes to cancer pain through PAR_2_ on neurons.

## 1. Introduction

Cathepsin S is a lysosomal cysteine protease that degrades proteins along the endocytic pathway, including the invariant chain necessary for MHC class II antigen processing and presentation [1]. Of the 11 human cathepsins, most require an acidic environment, such as the interior of lysosomes, to be active. Cathepsin S is distinct because it is active at a wide pH range of 4.5–8.0 [2]. Because of the capacity of cathepsin S to function at a neutral pH, it is active in the extracellular environment, where it degrades extracellular matrix proteins. The dysregulated expression and activity of proteases contributes to pathologic conditions including inflammation, pain and cancer. For most proteases, inhibition as a therapy is limited by the wide tissue expression of proteases and resultant side effects; for this reason, treatment with matrix metalloproteinase inhibitors is highly limited [3]. Cathepsin S, on the other hand, has limited tissue expression, primarily in antigen presenting cells in the lymph and spleen [4,5]. Thus, cathepsin S is an attractive target, and cathepsin S inhibitors have been tested in clinical trials (NCT 00425321, 01515358); the inhibitors exhibit good safety profiles [6].

Human cancers, including prostate, gastrointestinal (gastric, colorectal), lung, and brain tumors (astrocytoma, glioblastoma) upregulate cathepsin S [7,8,9]. Microdialysis, an extracellular fluid sampling approach originally designed to collect neurotransmitters, confirmed that cathepsin S was present in the extracellular space of brain tumors [10]. In the setting of cancer, both tumor cells and tumor-associated macrophages produce cathepsin S [11]. Cathepsin S contributes to several hallmarks of cancer including proliferation, angiogenesis, invasion and metastasis [8,12,13,14]. Mice in which the SV40 T antigen (Tag) transgene is controlled and driven by the rat insulin II promoter (RIP1-Tag2) spontaneously develop pancreatic beta-cell islet carcinoma. This mouse model was crossed with a cathepsin S null mouse, and cathepsin S was shown to regulate invasion and angiogenesis [11,15]. In a separate study, the combined depletion of cathepsin S in the cancer and macrophages was required to reduce the metastasis of breast cancer to the brain [16]. More recently, cathepsin S has been shown to mediate autophagy and apoptosis in human oral cancer cell lines through p38 MAPK/JNK signaling pathways [17]. 

In addition to carcinogenesis, cathepsin S produces pain. Cathepsin S cleaves protease-activated receptor-2 (PAR_2_), a G protein-coupled receptor (GPCR) expressed on peripheral nociceptors that mediates neurogenic inflammation and pain [18]. Peripheral nerve injury leads to cathepsin S upregulation and secretion from microglia. The cathepsin S inhibitor—morpholinurea-leucine-homophenylalanine-vinyl phenyl sulfone (LHVS)—reverses neuropathic pain [19]. Although the role of cathepsin S in carcinogenesis and pain has been studied, the question of whether cathepsin S contributes to cancer pain has not been answered. We hypothesized that cathepsin S released by oral cancer and/or associated macrophages in the oral cancer microenvironment generates cancer pain through the activation of PAR_2_. To test our hypothesis, we measured cathepsin S expression and activity in human and mouse oral cancers, immunolocalized cathepsin S in cancer cells and macrophages in human oral cancers, and used a series of mouse models, including a model in which the gene for PAR_2_ is selectively deleted on sensory neurons, to test whether cathepsin S mediated oral cancer pain.

## 2. Materials and Methods

### 2.1. Oral SCC Patients

Patients were screened and enrolled through the NYU Oral Cancer Center after consent was obtained. Detailed demographic information (age, sex, ethnicity, cancer location, primary tumor stage, and evidence of metastasis) was collected. Self-reported oral cancer pain was measured with the University of California Oral Cancer Pain Questionnaire prior to surgery. The instrument consists of 8 questions; question 7 addresses mechanical sensitivity [20,21]. The questionnaire uses a visual analog scale for each question, which ranges from 0 to 100. During surgical resection, tumor and matched normal oral mucosa specimens were collected (normal was harvested at an anatomically matched contralateral site). Specimens were frozen in liquid nitrogen and maintained at −80 °C. For one patient with tongue cancer, a portion of the lingual nerve innervating the side of the tongue affected by cancer was harvested as part of the resection specimen. In the same patient, the contralateral lingual nerve innervating the unaffected tongue was harvested. The lingual nerves were immersed in 10% neutral buffered saline for 24 h at 4 °C. The nerves were then washed 3 times with PBS and kept in 70% ethanol. The nerves were embedded in paraffin and sectioned at 5 µm. The Committee on Human Research at NYU Langone Medical Center approved human studies (10-01261, 15 September 2020).

### 2.2. Mice

Female and male wild-type (WT) C57BL/6J (stock number 000664) and NU/J *Foxn1*^nu^ athymic (stock number 002019), which were four to eight weeks old, were obtained from The Jackson Laboratory at Bar Harbor, ME. To delete *Par_2_* in peripheral neurons, *F2rl1* conditional knock-out (KO) C57BL/6 mice generated by genOway in Lyon, France, as described in [22], were crossed with mice expressing Cre recombinase under the control of the *Scn10a* gene promoter (B6.129-Scn10a^tm2(cre)Jnw^/H [23]. The animals were kept and bred in specific-pathogen-free rooms in the Animal Center of NYU College of Dentistry under the following conditions: a 12 h/12 h light/dark cycle, constant temperature of 22 ± 2°C and 60 ± 10% humidity. They received food and water ad libitum. The NYU Institutional Animal Care and Use Committee approved our studies in mice.

### 2.3. Tongue Xenograft Cancer Model

We generated the orthotopic xenograft tongue cancer model on the NU/J *Foxn1*^nu^ athymic mice of 4 to 6 weeks old by injecting 1 × 10^5^ HSC-3 (human tongue oral SCC cell line, cell number JCRB0623, from the Japanese Collection of Research Bioresources Cell Bank) into the tongue. The HSC-3 cells were suspended in 20 µL vehicle (1:1 mixture of DMEM and Matrigel; Corning, Ref. #354234). The mice were anesthetized by 1% isoflurane in 1L per minute medical oxygen during the injection. Two weeks after inoculation, xenografted tongues were collected and snap frozen for protein analysis.

### 2.4. Analysis of Total and Active Cathepsin S in Human Oral Cancer and Mouse Oral Cancer Tissues

Snap frozen human and murine tissues were sonicated in PBS, pH 7.4 (10 µL/mg tissue). Solids were cleared by centrifugation and protein concentration was measured by the BCA assay (Pierce). Protein was diluted in PBS (50 µg/20 µL buffer), and the cathepsin S-selective activity-based probe, BMV157, was added from a 100× DMSO stock (1 µM final) [24]. Samples were incubated at 37 °C for 30 min and the reaction was quenched with 5× sample buffer (200 mM Tris-Cl [pH 6.8], 8% SDS, 0.04% bromophenol blue, 5% β-mercaptoethanol, and 40% glycerol). Protein was resolved on a 15% polyacrylamide gel under reducing conditions. BMV157 binding was detected by scanning the gel for Cy5 fluorescence using a Typhoon 5 (GE Healthcare, Chicago, IL, USA). Proteins were transferred to nitrocellulose membranes for immunoblotting. The following antibodies were diluted in 50% LI-COR blocking buffer and 50% PBS-T containing 0.05% Tween-20: goat anti-human cathepsin S antibody (1:500, AF1183, lot # ICO0818121, R&D, Minneapolis, MN, USA); donkey-anti goat-HRP (1:10,000, A15999, Invitrogen, Waltham, MA, USA). Clarity Western ECL Substrate (Bio-Rad, CA, USA) was used for detection.

### 2.5. Quantification of Cathepsin S Activity in Oral SCC Cells and Dysplastic Oral Keratinocytes

HSC-3 or DOK cells were seeded in 6-well plates and switched to serum-free medium when 80% confluency was reached. After 17 h, cells were live labeled with the cathepsin S-selective probe BMV157 or the pan cathepsin S probe BMV109 (1 µM, 0.1% DMSO) for 7 h [25,26]. Cells were washed with PBS, lysed on ice in PBS containing 0.1% Triton X-100. Solids were cleared by centrifugation and proteins were solubilized in sample buffer. Total protein (~60 µg) was resolved by SDS-PAGE. Gels were scanned for Cy5 fluorescence and proteins were transferred to nitrocellulose membranes for cathepsin S immunoblotting as described above. The rabbit anti-actin antibody (1:10,000, A5060, lot #068M4870V, Sigma Aldrich, St. Louis, MO, USA) was detected with goat anti-Rabbit-IRDye 800CW (1:10,000, 926-32211, LI-COR) on the Typhoon 5.

### 2.6. Multiplex Immunostaining of Human Oral SCC Tissue and Adjacent Normal Mucosa 

Five-micron paraffin-embedded human oral SCC tongue or adjacent normal sections were stained either with H&E or with Akoya Biosciences^®^ Opal™ multiplex automation kit reagents on a Leica BondRX^®^ autostainer, according to manufacturers’ instructions. All slides underwent sequential epitope retrieval with either Leica Biosystems epitope retrieval 1 (ER1, citrate based, pH 6.0, Cat. AR9961) or epitope retrieval 2 solution (ER2, EDTA based, pH 9.0, Cat. AR9640), primary and secondary antibody incubation and tyramide signal amplification (TSA) with Opal^®^ fluorophores Op480, Op570 and Op690. The primary antibodies against human cathepsin S (1:100, Cat # SC-74429, Santa Cruz, Dallas, TX, USA), CD68 (1:100, Cat # GA609, Agilent, Santa Clara, CA, USA) and CK5 (1:1200 dilution, Cat # PRB-160P, Biolegend, San Diego, CA, USA), and the horse radish peroxidase-coupled secondary antibodies (Cat # ARH1001, Akoya HRP Polymer) were removed during sequential heat retrieval steps while fluorophores remained covalently attached to the epitope. ER1 was used for 60 min for the epitope retrieval of the antibody against human cathepsin S. For antibodies against CD68 and CK5, epitope retrieval was performed with ER2 for 20 min. Sections were counter-stained with DAPI. Semi-automated whole slide scanning was performed on a Vectra^®^ Polaris multispectral imaging system and the images were visualized with Akoya Phenochart software.

### 2.7. Measurement of F2RL1 mRNA, with RNAscope^®^, in Human Tongue Cancer and Contralateral Normal Tongue in the Same Patient

RNAScope^®^ chromogenic in situ hybridization (Advanced Cell Diagnostics by Bio-Techne, MN, USA) was performed according to the manufacturer’s pretreatment protocol for fresh-frozen tissue. Samples were hybridized with RNAscope^®^ probe Hs-*F2RL1* and signals were visualized by RNAscope^®^ 2.5 HD Assay RED and counterstained with hematoxylin. All the slides were scanned at the maximum available magnification and stored as digital high-resolution images. Chromogenic dots were quantified with NIH ImageJ software. Five randomly chosen fields at 40× magnification were counted by a blinded investigator.

### 2.8. Measurement of PAR_2_ Protein, with Immunohistochemistry, in the Human Lingual Nerve Innervating Tongue Cancer, Compared to the Lingual Nerve Innervating Contralateral, Unaffected Tongue

After deparaffinization, we performed antigen retrieval with sodium citrate buffer at pH 6, 82 °C for 20 min. The samples were kept at room temperature for 30 min and then washed three times with PBS. The samples were treated with 0.1% Triton X-100 in PBS for 5 min, followed by a blocking step with 3% bovine serum albumin (BSA) in PBS for 1 h at room temperature. The rabbit anti-PGP9.5 antibody (concentration 2 µg/mL, PB9840, Boster Biological Technology,) and mouse anti-PAR_2_ antibody (1:50 in 3% BSA in PBS, SAM11, sc-13504, Santa Cruz Biotechnology) were applied to the sections overnight at 4 °C. The samples were washed 3 times with PBS. The sections were then treated with secondary antibodies, which included goat anti-rabbit 594, (1:300 in 3% BSA in PBS, A32740, Invitrogen), and goat anti-mouse 488 (1:300 in 3% BSA in PBS, A11029, Invitrogen) for 2.5 h at room temperature. The sections were washed three times with PBS, treated with Prolong™ Diamond Antifade Mountant (P36965, Invitrogen) and coverslips were applied. Images were captured with a laser confocal microscope Carl Zeiss LSM 700 within 3 days after staining. For the negative control staining of PAR_2_, we performed immunohistochemical staining on HSC-3 cancer cells in which *F2RL1* was deleted with CRISPR/Cas9. Deletion of *F2RL1* resulted in the lack of PAR_2_ signal. PAR_2_ signal intensity was quantified in the lingual nerve innervating the tongue cancer (*n* = 300 axons), and the lingual nerve innervating contralateral normal tongue (*n* = 275 axons) with NIH ImageJ software by a blinded investigator.

### 2.9. FLAG Imaging

HEK-FLAG-PAR_2_ cells have previously been characterized [27]. HEK-FLAG-PAR_2_ cells were plated on poly-D-lysine-coated 12 mm glass coverslips and treated as described previously [27]. Cells were exposed to HSC-3 supernatant and fixed with 4% paraformaldehyde. Cells were washed with PBS two times and blocked with PBS + 0.3% saponin + 3% NHS for 1 h at room temperature. Cells were incubated with mouse anti-FLAG (1:500, #8146, Cell Signaling) overnight at 4 °C. Cells were washed three times in PBS and incubated with donkey anti-mouse Alexa 488 (1:1000, A32766, Thermo Fisher Scientific) for 1 h at room temperature. Cells were incubated with DAPI for 5 min (1 μM) in saline, washed 4x with PBS, and mounted with the ProLong Glass hard-set mounting medium (Thermo Fisher Scientific, Waltham, MA, USA). Cells were imaged in a Leica SP8 laser scanning confocal microscope. Micrographs were processed using ImageJ and Adobe Illustrator.

### 2.10. On-Cell Westerns Assay

FLAG-PAR_2_ cells were plated on poly-D-lysine-coated 96-well plates (30,000 cells/well) incubated for 48 h. HSC-3 supernatant was thawed and warmed to 37 °C and was pretreated for 30 min with either vehicle (DMSO), LY3000328 (1 μM) or cystatin C (1 μM) for 30 min. Cells were washed two times in HBSS (pH 7.4) and incubated with 100 μL of either *CTSS*^+/+^ HSC-3 supernatant, *CTSS*^−/−^ HSC-3 supernatant, trypsin (10 nM), cathepsin S (100 nM), or HSC-3 supernatant pretreated with either LY3000328 or cystatin C. Cells were incubated for 30 min at 37 °C, washed 1× in HBSS and fixed with 4% paraformaldehyde in PBS for 20 min on ice. Cells were washed two times in PBS and incubated with blocking buffer (PBS + 3% normal horse serum, NHS, Cat. # 31874 Thermo Fisher Scientific Waltham, MA, USA) for 1 h at room temperature. Cells were incubated with mouse anti-FLAG antibody (1:500, Cell Signaling, Boston, MA, USA) in PBS + 1% NHS overnight at 4° C. Cells were washed two times in PBS and incubated with donkey anti-mouse Alexa 790 (1:1000, A11371, Thermo Fisher) in PBS + 1% NHS for 1 h at room temperature. Cells were washed in PBS and incubated with SYTO™ 82 Orange (1 μM, Thermo Fisher) in saline for 30 min. Cells were washed two times in PBS and imaged on an Amersham Typhoon imaging system (GE, Pittsburg, PA, USA). FLAG immunofluorescent labeling was quantified using ImageJ and was normalized to the nuclear fluorescent intensity to correct for cell loss.

### 2.11. Transfection and Clonal Isolation

HSC-3 cells plated in a 6-well plate at a density of 2 × 10^5^ cells per 3 mL of DMEM supplemented with 10% fetal bovine serum (FBS) were incubated for 24 h. To generate *CTSS* knockout cells, 1 μg of *CTSS* CRISPR/Cas9 KO plasmid (cat# sc-417407, Santa Cruz Biotechnologies, Dallas, TX, USA), 1 μg of *CTSS* HDR plasmid (cat# sc-41407-HDR, Santa Cruz Biotechnology) and modified Tat (1 mM) were combined in a 5% glucose solution with a final volume of 60 μL/well. The solution was mixed for 5 s. After the mixture was incubated at room temperature for 30 min, the cap of the sample tube was removed to expose the solution to the air. The sample was then vigorously shaken for 90 min. Eight μL of FuGENE HD transfection reagent (cat# E2311, Promega, Madison, WI, USA) was added and then the solution was incubated at room temperature for 15 min. Sixty μL complexes (containing 2 μg DNA and FuGENE) were placed in the wells of a 6-well plate. The plate was gently mixed and incubated for 48 h. Following incubation, the medium was replaced with RPMI complete medium with 500 ng/mL puromycin dihydrochloride (Santa Cruz Biotechnology) every 2 days to eliminate wild-type puromycin-sensitive cells. After puromycin selection, single-cell clones were isolated by limiting dilution into 96-well plates. Clones were expanded and transferred to larger plates as the individual clones reached confluence. Clones were genotyped by RT-PCR. Finally, cathepsin S expression was determined by ELISA assay. For controls, 1 μg of control CRISPR/Cas9 plasmid (cat# sc-4148922, Santa Cruz Biotechnology) was used.

### 2.12. RT-PCR

For reverse-transcription polymerase chain reaction (RT-PCR), total RNA was isolated using TRIzol reagent (Invitrogen). RNA concentration was measured by NanoDrop ND-1000 Spectrophotometer (NanoDrop Technologies, Wilmington, DE, USA) and cDNA was synthesized from a total of 1 μg RNA using QuantiTect^®^ Quantiscript reverse-transcriptase and RT Primer Mix (Qiagen, Hilden, Germany), according to the manufacturer’s protocol. RT-PCR was performed using Taq PCR Master Mix Kit (Qiagen), according to standard protocols. The sequence of beta actin (*ACTB*) and cathepsin S (*CTSS*) gene-specific primers were as follows: *ACTB* (5′-CATGTACGTTGCTATCCAGGC-3′(sense) and 5′- CTCCTTAATGTCACGCACGAT-3′(antisense); product size 250bp) and *CTSS* (5′- TGACAACGGCTTTCCAGTACA-3′(sense), 5′- GGCAGCACGATATTTTGAGTCAT-3′(antisense)]; product size 113 bp). PCR products were analyzed with 1% agarose gel electrophoresis.

### 2.13. Cathepsin S Quantification by ELISA

HSC-3 or OSC-20 cells were plated in a 12-well plate (2 × 10^5^ cells/well) and were incubated for 24 h. Medium was removed, cells were washed with 3 mL PBS without Ca^2+^ and Mg^2+^, and DMEM (500 µL) was added to each well. After 48 h, the medium was collected and centrifuged (1000 rpm, 5 min, 4 °C). The pellet was discarded. Total cellular protein was collected from HSC-3 and OSC-20 cell lines (cell number JCRB0197, was from Japanese Collection of Research Bioresources Cell Bank) using standard RIPA lysis buffer (500 µL/well). Cathepsin S concentration was quantified by enzyme-linked immunosorbent assay (ELISA) for human cathepsin S (Cat#: ab155427, Abcam, Cambridge, UK). A standard curve was constructed before each experiment. Protein samples were diluted in dilution buffer and 100 μL was added to each well. The plate was incubated overnight at 4 °C and washed four times with washing solution. The diluted biotinylated anti-human cathepsin S antibody (100 μL) was added to each well and incubated for 1 hour at room temperature. Following the washing steps, 100 μL of 1× HRP–streptavidin solution was added to each well and the plate was kept in a dark environment at room temperature for 45 minutes. After incubation, the plate was washed and TMB substrate reagent (Cat#: ab210902, Abcam) was added to each well. Finally, the reaction was stopped by the addition of 50 μL of stop solution. The absorbance was read at 450 nm in a microtiter plate reader immediately. The absorbance was measured using a 450 nm filter on GloMax^®^-Multi Microplate Multimode Reader (Promega).

### 2.14. Facial Mechanical Nociception

The rat facial mechanical nociception assay was modified for mice [27,28]. Two weeks prior to the assay, mice were acclimated for 1 h in the testing room every other day. In ascending order, von Frey filaments ranging from 0.008 to 4 g force (11 filaments in total) were used to measure withdrawal responses to mechanical stimulation of the left cheek. Each fiber was applied once to the cheek, defined by the area between the nose and the ear, below the eye. If the mouse was moving or the response was unclear to the researcher, the same von Frey filament was reapplied to the same area of the cheek 10 s after the first stimulus or until the mouse stopped moving. A 5 min interval was set between the applications of von Frey filaments of different intensities. The facial nociception score was reported as a numerical average of the 11 responses in the response categories as we reported [22]. In some experiments, gavage treatment of LY3000328 (cathepsin S inhibitor) or the subcutaneous injection of cystatin C (endogenous cathepsin inhibitor) into the cheek was administered for 2 or 1 h, respectively, before the facial mechanical nociception assay under 1% isoflurane in 1 L per minute medical oxygen.

### 2.15. Paw Xenograft Cancer Model

The paw xenograft cancer model permits the measurement of mechanical and thermal nociception in the paw. Baseline mechanical and thermal withdrawal thresholds were measured prior to tumor cell inoculation. The left hind paw of NU/J *Foxn1*^nu^ athymic mice of 4 to 6 weeks old was injected with 1 × 10^5^ HSC-3 in 20 μL of DMEM and matrigel (1:1) [29,30]. Mechanical and thermal nociception assays were conducted at post inoculation weeks 1, 2, 3 and 4. In some experiments, on post inoculation week 4, LY3000328 (cathepsin S inhibitor, 30 mg·kg^−1^, 100 µL) was administered by oral gavage. Mechanical and thermal nociception assays were performed at 1, 3, 6, 12, and 24 h after LY3000328 treatment. At the end of the experiments, the cancer paw was collected and processed for histological hematoxylin and eosin (H&E) staining to confirm viable carcinoma.

### 2.16. Mechanical and Thermal Nociception in the Hind Paw

To assess mechanical nociception, mice were placed on a platform with a metal mesh floor and acclimated for 1 h. The paw withdrawal threshold was measured with von Frey filaments (Stoelting, Wood Dale, IL, USA) according to the up–down method for rats published by Chaplan et al. in 1994 with modifications for mice [31,32]. The withdrawal threshold was defined as the gram-force sufficient to elicit left hindpaw withdrawal. A positive response was recorded if the mice showed one of the following reactions: 1—quick paw withdrawal; 2—immediate flinch when the tip of the von Frey filament is removed; 3—digit extension; 4—paw lift and licking of the paw; 5—repeated flapping of the paw to the mesh; or 6—attempt to run to escape from the stimulation. In some tests, the withdrawal threshold for each animal was determined as the mean of 3 trials for each animal. The interval between two trials was 10 s. The cut-off value was 4 g to prevent the paw from mechanical injury. 

To assess thermal nociception in the paw, we used a thermal stimulator (IITC Life Sciences, Woodland Hills, CA, USA) [33]. Two weeks before the assay was performed, the mice were acclimated to the stimulator for one hour every other day. Mice were placed individually in a plastic chamber on a 25 °C glass surface. A radiant heat source was focused on the left hind paw and withdrawal latency was measured in seconds from the time the heat source started to project into the paw until the time the mice withdrew its paw. The outcome variable was the mean of 3 trials undertaken at intervals of 5 or more minutes. A 20 s cut-off latency was established to prevent heat injury of the paw.

### 2.17. Cancer Paw Volume Measurement

The paw xenograft model permits the measurement of cancer paw volume at the time of behavioral assessment. Prior to cancer cell inoculation and at 1, 2, 3 and 4 weeks after inoculation, a plethysmometer (cat #. II-520MR, World Precision Instruments, FL, USA) was used to measure the volume of cancer paw in anesthetized mice. Volumes were measured 3 times and reported as the mean.

### 2.18. Histologic Determination of the Area of Cancer in the Paw

Four weeks after the inoculation of cancer cells into the paw, the cancer paw was collected. The paw was defined as the area from the tip of the digits to the ankle crease. The paw was longitudinally sectioned through the thickest area of the tumor, which generated 2 cut surfaces within the largest area of cancer. The paw was then immersed in 10% neutral buffer formalin (NBF) for 48 h at 4 °C. NBF was then replaced with 10% ethylenediaminetetraacetic acid (EDTA) for paw decalcification. The EDTA solution was changed every 2 days for 14 days and then replaced with 70% Et-OH for 3 days prior to paraffin embedding. One section was made on each cut surface of the cancer paw for H&E histological staining. The H&E image was scanned on a Hamamatsu Nanozoomer. The area of cancer was determined as the average area from the 2 sections as quantified by the NIH ImageJ^®^ software. Two independent researchers quantified the area of cancer and the average was recorded.

### 2.19. Statistics

GraphPad Prism 7 and 8 (GraphPad Software, Inc., San Diego, CA, USA) was used for the statistical analysis. Results were expressed as mean ± standard error (SEM). For cell-based assays, triplicate measurements were made; differences were evaluated by one- or two-way repeated measures ANOVA and Dunnett’s or Tukey’s multiple comparisons test. One-way ANOVA and Tukey’s or Sidak’s multiple comparisons and Student’s *t*-test were used for in vivo behavioral experiments. An independent samples Student’s *t*-test was used to compare values between 2 groups. The relationship between participant pain scores, as measured by the University of California San Francisco Oral Cancer Pain Questionnaire, and cathepsin S activity was analyzed using Spearman correlation.

## 3. Results

### 3.1. Cathepsin S Activity and Expression in Human Oral Cancers

To determine whether cathepsin S is activated in oral SCCs, we collected oral SCC specimens and matched normal oral mucosa from seven patients (Table 1). Specimens were incubated with a fluorescently quenched activity-based probe (BMV157) selective for cathepsin S. A BMV157-labeled species of 25 kDa was activated in oral SCC versus normal tissue (Figure 1a). Oral SCC showed increased expression and activity of cathepsin S relative to matched normal tissue in all patients (Figure 1b–e). All participants reported pain. Self-reported mechanical sensitivity tended to increase with cathepsin S activity in the tumor (r_s_ = 0.78, *p* = 0.041)

We used multiplex immunostaining to localize cathepsin S in human oral SCC patients (Table 2). Adjacent normal tissue in one of the oral SCC patients (Table 2, Patient A) served as the control. Cathepsin S was expressed in CD68^+^ macrophages in human oral SCC tissue (Figure 2). Cathepsin-S expression was extremely low in an adjacent normal tissue section from an oral SCC patient (Figure 2). In separate tissue sections from the same patients (Table 2), cathepsin S was identified within keratin-positive cells (CK5^+^) indicating that cathepsin S was expressed by both tumor cells and macrophages within the oral SCC microenvironment (Figure 3).

### 3.2. F2RL1 mRNA in Human Tongue Cancer Compared to Contralateral Normal Tongue, and PAR_2_ Protein Expression in the Lingual Nerve Innervating the Tongue Cancer Compared to the Lingual Nerve Innervating the Contralateral Unaffected Tongue

We measured *F2RL1* mRNA, with RNAscope^®^ in situ hybridization, in a human tongue cancer and compared it to contralateral unaffected tongue in the same patient (i.e., matched) (Figure 4a,b). The level of *F2RL1* mRNA was five times higher in the tongue cancer than in the matched tongue tissue (3.0 ± 0.4 in cancer tongue versus 0.6 ± 0.2 in normal tongue, Figure 4b). We measured PAR_2_ protein, with immunohistochemistry, in the lingual nerve innervating the tongue cancer (Figure 4d) and compared it to the lingual nerve innervating the contralateral unaffected tongue (Figure 4c). PAR_2_ protein was significantly higher in the lingual nerve innervating the tongue cancer than in the lingual nerve innervating the contralateral tongue (Figure 4e). 

### 3.3. Cathepsin S Activity in Mouse Oral Cancers

Using BMV157, we measured cathepsin S activity in the mouse orthotopic xenograft model in which HSC-3 human oral SCC cells were inoculated into the tongue. Compared to normal mouse tongues, HSC-3 orthotopic xenografts exhibited increased cathepsin S activity and increased total levels of immunoreactive cathepsin S (Figure 5a–c). Thus, the xenograft model mimics our findings in human SCC.

### 3.4. Cathepsin S Activity in a Human Oral Cancer Cell Line Compared to a Human Dysplastic Oral Keratinocyte Line

We measured cathepsin S activity in HSC-3 cells and dysplastic oral keratinocytes (DOK, non-cancer cell line, cell number, 94122104, was from SIGMA-ALDRICH) with BMV157, the cathepsin S-selective probe, and BMV109, a pan cathepsin probe. Cathepsin S activity was higher in HSC-3 than DOK, as measured by both probes (Figure 6a–c). Likewise, total cathepsin S levels were increased in HSC-3 cells compared to DOKs (Figure 6a,d). We also used ELISA to measure the intracellular and secreted cathepsin S in HSC-3 cells as well as a second human oral cancer line, OSC-20 (Figure 6e).

### 3.5. Cathepsin S from Human Oral Cancer Cell Lines Cleaves PAR_2_

To determine whether cathepsin S in cancer supernatant cleaves the exodomain of PAR_2_, FLAG-PAR_2_ plasmids were transiently transfected in HEK cells and PAR_2_ cleavage was measured with the On-Cell Western assay. These cells expressed PAR_2_ with an extracellular N-terminal FLAG epitope that is upstream of the site of cathepsin S cleavage (Figure 7a). In cells treated with the negative control vehicle, the FLAG epitope was intact, thus it was visualized on the plasma membrane using immunohistochemistry with anti-FLAG antibody (Figure 7b, left image, green color). In contrast, cathepsin S (100 nM for 30 min) cleaved the exodomain of PAR_2_, and caused a loss of surface FLAG immunoreactivity (Figure 7b, right image, green color). The result indicates that cathepsin S cleaved PAR_2_ and removed the FLAG epitope.

The On-Cell Western assay was also used to quantify the removal of the FLAG epitope. In cathepsin S-incubated cells (Cat S, 100 nM, 30 min, 37 °C), FLAG immunoreactivity was reduced by 66.02% (1.94 ± 0.17 in vehicle-treated cells versus 0.66 ± 0.14 in Cat S-treated cells). Similarly, the FLAG immunoreactivity was reduced by 42.83% when supernatant from the HSC-3 *CTSS^+/+^* was applied to cells in comparison to the supernatant from the HSC-3 *CTSS*^−/−^ (1.55 ± 0.16 in *CTSS*^−/−^ versus 0.90 ± 0.17 in *CTSS^+/+^*) (Figure 7c). LY3 and cystatin C inhibit PAR_2_ cleavage by *CTSS^+/+^* cells (Figure 7c). Collectively, these results indicated that cathepsin S from human cancer cell lines cleaves membrane PAR_2_.

### 3.6. Cathepsin S Causes Orofacial Nociception That Is Neuronal PAR_2_ Dependent

We previously demonstrated that cathepsin S produces nociceptive behavior through the cleavage of PAR_2_ when injected into tissue innervated by dorsal root ganglia [34]; however, the nociceptive effect of cathepsin S in the trigeminal system has not been demonstrated. To study the nociceptive effect of cathepsin S on orofacial nociception, we injected cathepsin S into the cheek that is innervated by trigerminal neurons (Figure 8a,d,g). The facial von Frey nociception assay was conducted before and at 1, 3, 6, 12 and 24 h after cathepsin S injection into the cheek to monitor the cathepsin S-evoked nociceptive behavior. We found that cathepsin S increased the facial nociception score compared to the vehicle control demonstrating that cathepsin S induces orofacial nociception (Figure 8b,e,h). To confirm that the nociceptive effect was dependent on the proteolytic activity of cathepsin S, we used a specific cathepsin S inhibitor, LY3000328, (Figure 8a–c) or an endogenous cathepsin S inhibitor, cystatin C (Figure 8d–f). Both cathepsin S inhibitors (Figure 8b,c,e,f) reversed the nociceptive effect of cathepsin S. We then questioned whether the nociceptive effect of cathepsin S was dependent on neuronal PAR_2_. We injected cathepsin S into the cheek of WT C57BL/6J mice and Par_2_Na_v_1.8 KO mice which lack the gene for PAR_2_ in Na_v_1.8-positive neurons (Figure 8g). The Par_2_Na_v_1.8 KO mice exhibited a 56.0% reduction in the facial nociception score at 1 h after cathepsin S injection (Figure 8h), and a 57.3% reduction over the course of 24 h after cathepsin S injection (Figure 8i).

### 3.7. Deletion of CTSS with CRISPR/Cas9 in HSC-3 Reduces Nociception, but Not Tumor Volume in the Xenograft Model

Having demonstrated that purified recombinant cathepsin S can invoke facial nociception in vivo, we then investigated the ability of tumor cell-derived cathepsin S to provoke oral cancer pain. We deleted *CTSS* in HSC-3 by CRISPR/Cas9. RT-PCR semi-quantitative analysis of *CTSS* gene expression in HSC-3 cells confirmed the lack of *CTSS* in HSC-3 *CTSS*^−/−^ (Figure 9a). An ELISA assay on an HSC-3 culture supernatant and cell lysate confirmed the lack of cathepsin S protein in HSC-3 *CTSS*^−/−^ (Figure 9b). To monitor the development of cancer-induced mechanical allodynia and thermal hyperalgesia, we used the paw von Frey nociception assay (Figure 9c,d) and the Hargreaves thermal assay (Figure 9c,e), respectively. Measurements were taken prior to the inoculation of HSC-3 cells in the paw (baseline) and weekly thereafter for four weeks. Xenografts generated from cathepsin S-deficient HSC-3 cells (*CTSS*^−/−^CRISPR/Cas9) provoked reduced mechanical allodynia and thermal hyperalgesia relative to xenografts generated from wild-type HSC-3 cells (untreated naïve cells or cells treated with random guide RNA) (Figure 9c,d,e). Cancer mice generated with HSC-3 *CTSS*^−/−^ showed a 58.7% reduction in mechanical allodynia (Figure 9d) and an 87.0% reduction in thermal hyperalgesia (Figure 9e) compared to cancer mice generated with HSC-3 treated with random CRISPR/Cas9 at 4 weeks after inoculation. 

We tested whether the deletion of *CTSS* in HSC-3 with CRISPR/Cas9 altered tumor proliferation as measured by tumor volume. The cancer paw volume was measured before (BL) and every week until 4 weeks after HSC-3 inoculation (Figure 9f). The cancer paw volume was not different between mice generated with HSC-3 *CTSS*^−/−^ or HSC-3 treated with random CRISPR/Cas9 or naïve HSC-3 cells (Figure 9g). We confirmed the cancer growth in the paw by histology and H&E staining (Figure 9h). The cancer area quantified by histology showed no difference between groups (Figure 9i).

### 3.8. The Cathepsin S Inhibitor Reduces Cancer Nociception, but Not Tumor Volume, in Cancer Mice Generated with Two Human Tongue Oral Cancer Cell Lines, HSC-3 and OSC-20

We tested whether two human oral cancer cell lines produced nociception through cathepsin S. We generated separate groups of xenograft mice with HSC-3 and OSC-20. Prior to and following paw inoculation, the mice were tested every week for 4 weeks with the paw von Frey nociception assay and the Hargreaves thermal assay to monitor the development of mechanical allodynia and thermal hyperalgesia, respectively (Figure 10a). The HSC-3 cancer mice developed nociception at 2 and 3 weeks (Figure 10b,c) while the OSC-20 cancer mice developed cancer nociception at 3 and 4 weeks (Figure 10e,f) after the cancer cells were inoculated into the paw. When the mice developed cancer nociception, a single dose of the cathepsin S inhibitor, LY3000328, was administered. The mice were then tested with the mechanical and thermal assays at 1, 3, 6, 12 and 24 h after LY3000328 was administered (Figure 10a). LY3000328 reduced mechanical cancer allodynia in HSC-3 and OSC-20 by 55.7% and 47.7%, respectively. LY3000328 reduced thermal cancer hyperalgesia in HSC-3 and OSC-20 by 73.6% and 56.3%, respectively. We measured the cancer paw volume in the mice before and every week until 4 weeks after the HSC-3 or OSC-20 cancer cells were inoculated into the paw. The cancer paw volume was similar between treatment groups and control groups (Figure 10d,g); LY3000328 did not alter paw volume after administration to the mice.

## 4. Discussion

The primary question addressed by the present study was whether cathepsin S produces cancer pain. We also sought to determine whether the putative nociceptive mechanism involves PAR_2_ on neurons; a role for PAR_2_ is likely as we previously demonstrated that PAR_2_ plays a central role regulating cancer pain [35,36,37]. Specifically, mice completely lacking PAR_2_ exhibit reduced cancer-associated allodynia and orofacial dysfunction [35,36]. Mice lacking PAR_2_ specifically in nociceptive neurons (Par_2_Na_v_1.8 mice) also exhibit attenuated nociception, indicating a central role for neuronal PAR_2_ in cancer pain [27]. Furthermore, we also demonstrated that PAR_2_-activating proteases (e.g., TMPRSS2 and legumain) contribute to cancer pain in a PAR_2_-dependent manner [27,37] 

We first characterized cathepsin S expression and activity in human oral squamous cell carcinoma tissues and patient-matched normal oral mucosa using cathepsin S-specific antibodies and BMV157, an activity-based probe that enables the assessment of the proteolytically active fraction of cathepsin S. Total and active cathepsin S were increased in cancer tissues compared to normal tissues. In breast, prostate, colorectal and metastatic brain cancers, cathepsin S is supplied by the tumor cells and associated stromal cells, predominantly macrophages [9,38,39]. In accordance with other cancers, we observed cathepsin S production by both keratin^+^ tumor cells and CD68^+^ macrophages of oral cancer specimens collected from four patients. Cathepsin S expression in adjacent normal tissue was minimal. We also demonstrated that cathepsin S activity is upregulated in immortalized human oral cancer cell lines compared to dysplastic oral keratinocytes. We demonstrated that cathepsin S is secreted by oral cancer cells, and that it cleaves human PAR_2_. The orthotopic xenografts of human oral cancer cells also exhibited increased cathepsin S activity compared to naïve mouse tongue tissue.

Furthermore, purified cathepsin S was able to invoke facial nociception in vivo, also in an activity-dependent and PAR_2_-dependent manner, suggesting that cathepsin S would be capable of provoking oral cancer pain.

We next investigated the ability of tumor-supplied cathepsin S to provoke oral cancer pain. The deletion of cathepsin S specifically from HSC-3 tumor cells resulted in a reduction of mechanical allodynia and thermal hyperalgesia (58.7% and 87.0%, respectively) produced by paw xenografts. In the future, complementary studies involving the inoculation of HSC-3 cells into wild-type and cathepsin S-deficient nude mice would shed light on the contribution of stromal (macrophage)-supplied cathepsin S to cancer pain. Ultimately, we showed that the administration of a single dose of a cathepsin S-selective inhibitor LY3000328, which targets both tumor and stromal-supplied cathepsin S, was able to attenuate cancer pain for at least 12 h in two independent xenograft models [16].

The treatment of oral cancer pain is challenging because cancers are heterogeneous and because oral cancers produce and secrete several nociceptive mediators. Nociceptive mediators produced and secreted by oral cancer include, but are not limited to, nerve growth factor, ATP, endothelin-1 and other proteases [30,40,41]. The cathepsin S/PAR_2_ axis is a strategic target for cancer pain. Cathepsin S is active in the extracellular environment of the cancer and associated neurons. Accordingly, it could be antagonized without cellular uptake, as opposed to other cathepsins, which are active in lysosomes. Furthermore, PAR_2_ activates several types of TRP channels, including TRPV4 [42], TRPV1 [43,44] and TRPA1 [45], which amplify the action of cathepsin S. As a result, nociceptors responsive to several algesic mediator types are sensitized and induce hyperexcitability and sustained nociception. In addition, PAR_2_ transactivates the epidermal growth factor receptor (EGFR), which also mediates nociception [46,47,48]. Thus, if we could prevent the activation of PAR_2_ by cathepsin S, we could potentially abrogate the sensitization of the receptors that mediate the nociceptive action of myriad mediators produced and secreted by cancers. A potential concern with the inhibition of cathepsin S is that antigen processing, which is critical to the immune response to cancers, might be affected. This argument is countered by the finding that cathepsin S is upregulated by several cancers. Moreover, to the degree that evolutionary mechanisms underlie tumor progression, the natural selection of tumor cells may be less likely to favor the overexpression of cathepsin S if proliferation were reduced. In fact, the inhibition or deletion of the gene for cathepsin S reverses several hallmarks of cancer [13]. Because of the previous studies reporting that cathepsin S mediates carcinogenesis, we measured the proliferation and tumor volume following the genetic deletion and pharmacologic antagonism of cathepsin S. The antinociceptive effect that we demonstrated could reflect reduced proliferation and decreased tumor burden. We found that neither the deletion of *CTSS* nor a single dose of the cathepsin S inhibitor, LY3000328, affected the tumor volume; therefore, the mechanism of preventing the expression or inhibiting cathepsin S is antinociceptive. The nociceptive mechanisms that we demonstrated in oral cancer (a notoriously painful malignancy) likely overlap with other cancer pain mechanisms, including bone cancer pain. Nociceptive mediators/mechanisms common to oral cancer and bone cancer include, but are not limited to, endothelin, NGF, and PAR_2_ [30,40,41,49,50]. Cathepsin S also might contribute to visceral pain associated with gastrointestinal cancers including pancreatic cancer. Cathepsins, including cathepsin S, also mediate pancreatitis [51]. Because cathepsin S inhibitors have been clinically used and show good safety profiles, we are considering a trial to test their efficacy for the management of cancer pain [6].

## 5. Conclusions

Cathepsin S, a lysosomal cysteine protease, is present and active in the oral cancer microenvironment. The protease cleaves protease-activated receptor-2 (PAR_2_) on neurons to contribute to the severe pain that is associated with oral cancer. In humans, cathepsin S activity correlates with reported pain. Genetic deletion of the gene for cathepsin S in a human oral cancer cell line reverses nociception in a mouse cancer model generated with the cancer cell line. We conclude that cathepsin S is responsible for oral cancer pain through PAR_2_ on neurons.

## Figures and Tables

**Figure 1 cancers-13-04697-f001:**
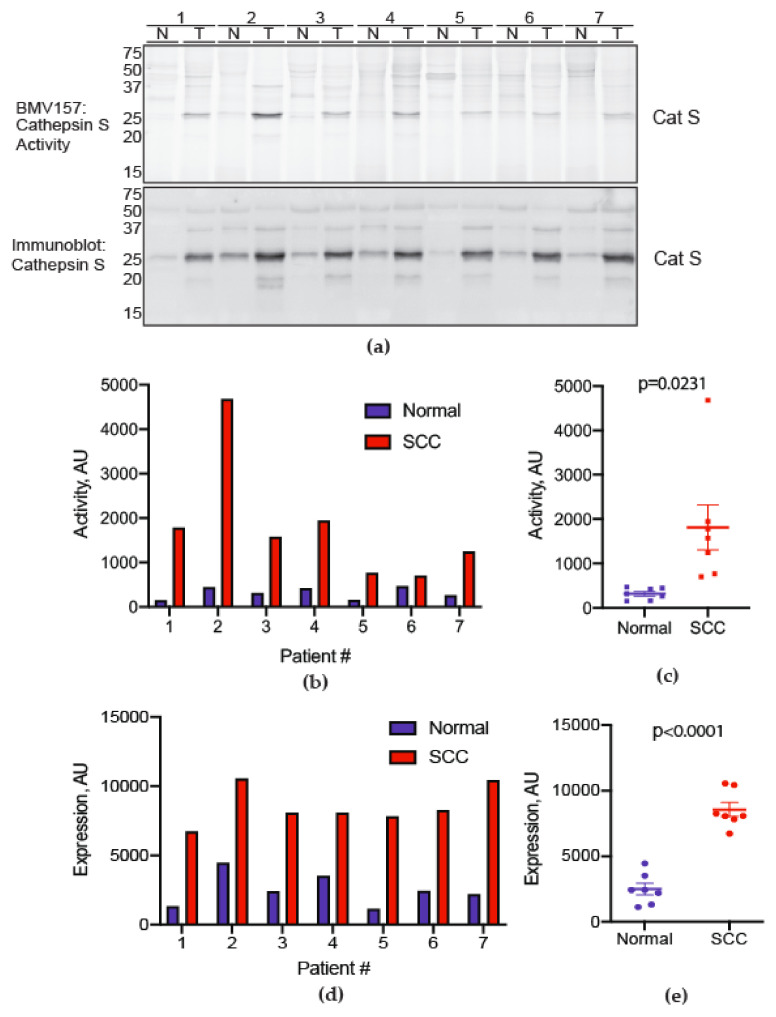
Cathepsin S activity is increased in human oral cancer compared to match normal tissue. (**a**) Active cathepsin S labeled by BMV157 (upper panel) as shown by in-gel fluorescence and total cathepsin S immunoreactivity (lower panel) in oral SCC biopsies (T) and patient-matched normal oral mucosa (N). The gel (upper panel) was transferred to nitrocellulose and immunoblotted for total cathepsin S levels (lower panel). The uncropped western blot figure is presented in Appendix A. (**b**) Densitometry of the 25 kDa species labeled by BMV157, displayed as individual values for all normal and oral SCC samples. (**c**) Average cathepsin S activity was higher in oral SCC samples (independent samples Student’s *t*-test). (**d**) Densitometry of the 25 kDa species labeled by BMV157, displayed as individual values for all normal and oral SCC samples. (**e**) Average total cathepsin S levels were higher in oral SCC samples (independent samples Student’s *t*-test).

**Figure 2 cancers-13-04697-f002:**
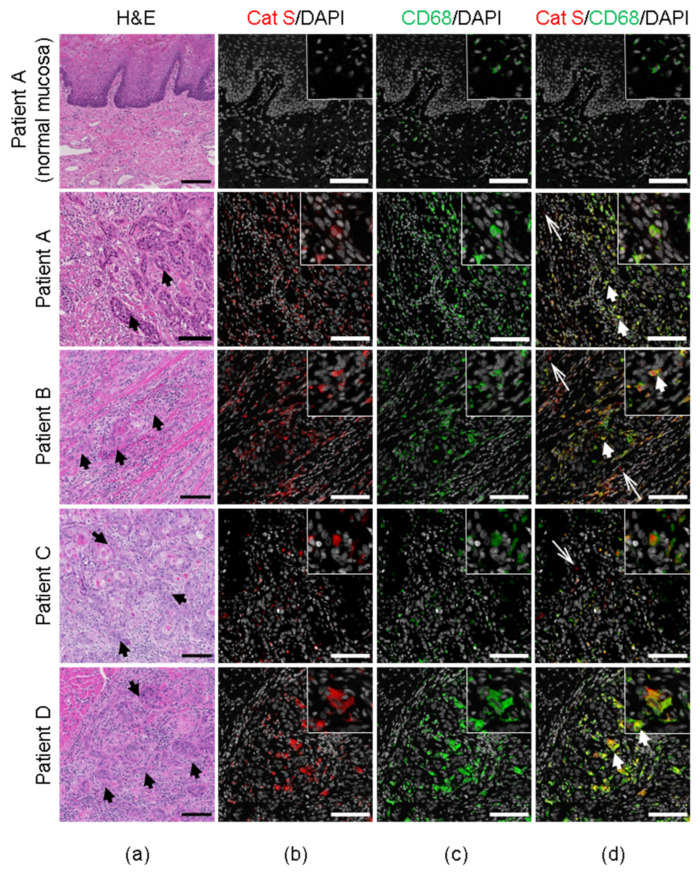
Localization of cathepsin S expression in macrophages associated with human oral cancers. (**a**) H&E staining of tongue tissue from four patients with tongue oral SCC, and adjacent normal tongue tissue section from one of the patients with oral SCC (Patient A) (Table 2). Black arrows in H&E sections indicate SCC. (**b**) Co-immunofluorescent staining of cathepsin S (Cat S, red) and DAPI (gray); (**c**) CD68^+^ macrophages (green) and DAPI (gray); (**d**) Cat S (red), CD68^+^ macrophages (green) and DAPI (gray). White thick arrows indicate cathepsin S in CD68^+^ macrophages; white thin arrows indicate cathepsin S in cancer microenvironment, but not in macrophages. Scale bars, 100 µm.

**Figure 3 cancers-13-04697-f003:**
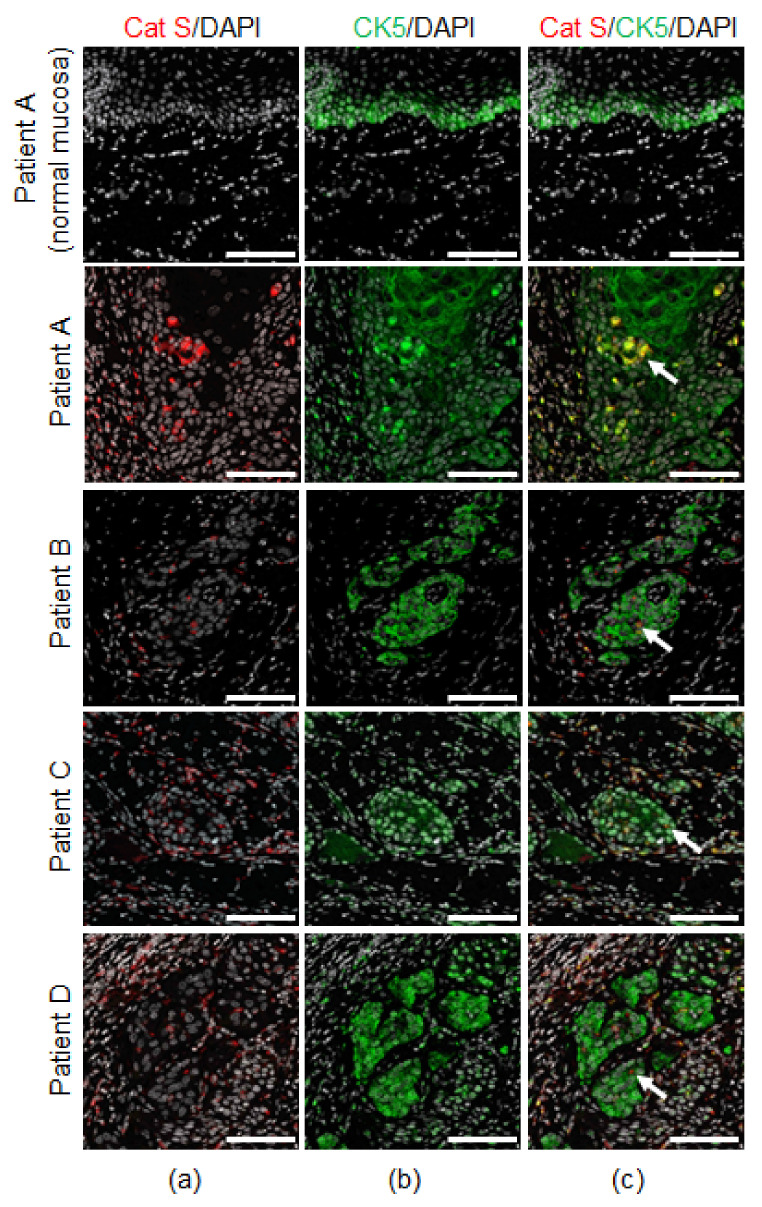
Localization of cathepsin S expression in human oral cancer cells. The immunofluorescent staining of four patients with oral SCC, and adjacent normal tongue tissue section from one of the patients with oral SCC (Table 2). (**a**) Co-immunofluorescent staining of cathepsin S (Cat S, red) and DAPI (gray); (**b**) cytokeratin for SCC cells (CK5, green) and DAPI (gray); (**c**) Cat S (red), CK5 (green) and DAPI (gray). White arrows indicate cathepsin S in SCC cells. Scale bars, 100 µm.

**Figure 4 cancers-13-04697-f004:**
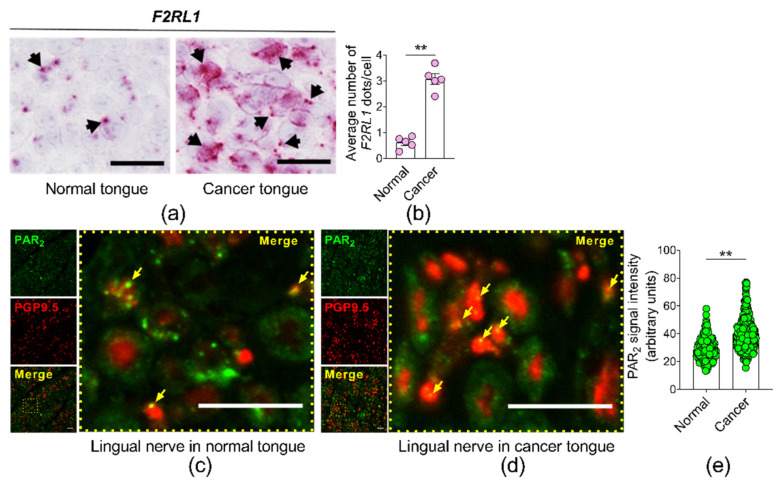
(**a**,**b**) *F2RL1* mRNA expression in a human cancer tongue is elevated compared to the contralateral unaffected normal tongue in the same patient. (**a**) Microscopy shows *F2RL1* expression in human cancer tongue (right panel) and contralateral unaffected normal tongue (left panel). Black arrows indicate *F2RL1* chromogenic dots. (**b**) The cancer tongue expressed approximately five times the amount of *F2RL1* mRNA expressed by the unaffected normal tongue. ** *p* < 0.01. One-way ANOVA. (**c**) PAR_2_ protein in human lingual nerve innervating contralateral unaffected normal tongue compared to (**d**) cancer tongue. The lingual nerves were counterstained with PGP9.5 (a neuronal marker). (**e**) PAR_2_ signal intensity in the lingual nerve innervating the cancer tongue (*n* = 300 axons) was higher than PAR_2_ signal intensity from lingual nerve innervating contralateral unaffected normal tongue (*n* = 275 axons). Arrows indicate PAR_2_ in axons. Scale bar, 10 μm. ** *p* < 0.01, Student’s *t*-test.

**Figure 5 cancers-13-04697-f005:**
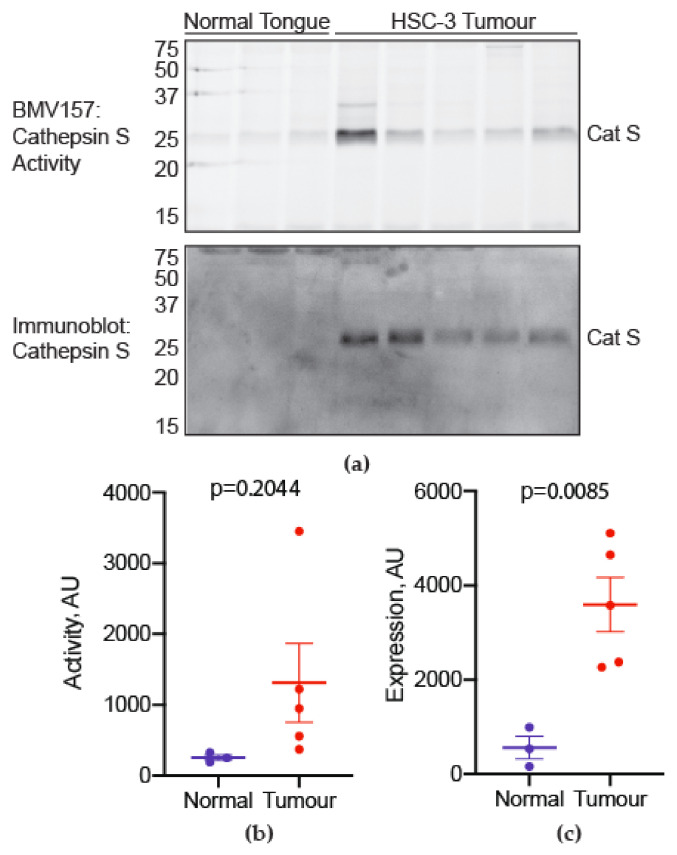
Cathepsin S activity is increased in the oral cancer of the mouse orthotopic xenograft model compared to mouse normal tongue. (**a**) Active cathepsin S labeled by BMV157 (upper panel) as shown by in-gel fluorescence and total cathepsin S immunoreactivity (lower panel). The gel (upper panel) was transferred to nitrocellulose and immunoblotted for total cathepsin S levels (lower panel). The uncropped western blot figure is presented in Appendix A. (**b**) Cathepsin S activity was higher in oral HSC-3 xenografts compared to normal tongues (unpaired Student’s *t*-test). (**c**) Cathepsin S expression was higher in HSC-3 xenografts compared to normal tongues (unpaired Student’s *t*-test).

**Figure 6 cancers-13-04697-f006:**
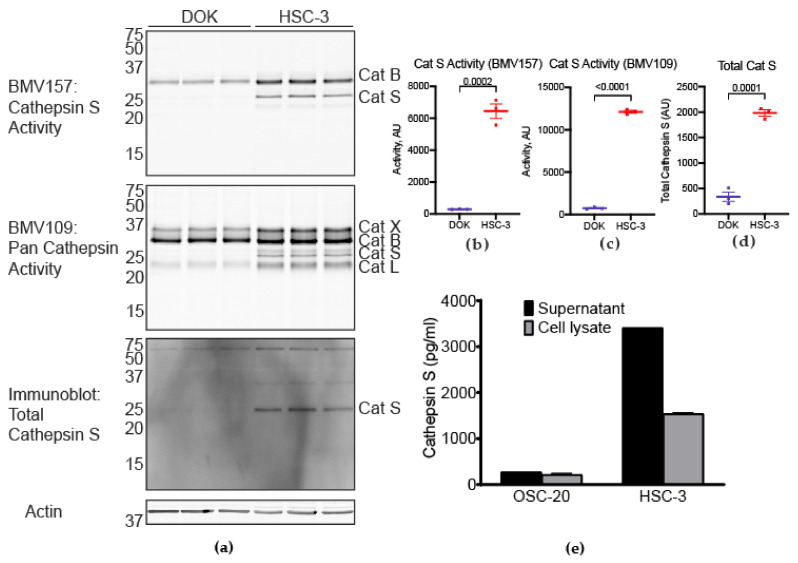
Cathepsin S activity and expression are increased in human oral cancer cells. Cathepsin S activity was compared between HSC-3 and the human dysplastic oral keratinocyte cell line, DOK. (**a**) Labeling of cathepsin S activity with BMV157 selective for cathepsin S in DOK and HSC-3, as shown by in-gel fluorescence; immunoblots confirmed cathepsin S. Labeling of pan cathepsin activity with BMV109 in DOK and HSC-3; immunoblots confirmed cathepsin X, B, S and L; cathepsin S activity is significantly higher in HSC-3 compared to DOK as measured by (**b**) BMV157, and (**c**) BMV 109. The uncropped western blot figure is presented in Appendix A. (**d**) Total cathepsin S activity in HSC-3 is significantly higher than in DOK. (**e**) ELISA assay to measure cathepsin S in supernatant and whole-cell lysate from HSC-3 and a second human oral squamous cell line, OSC-20.

**Figure 7 cancers-13-04697-f007:**
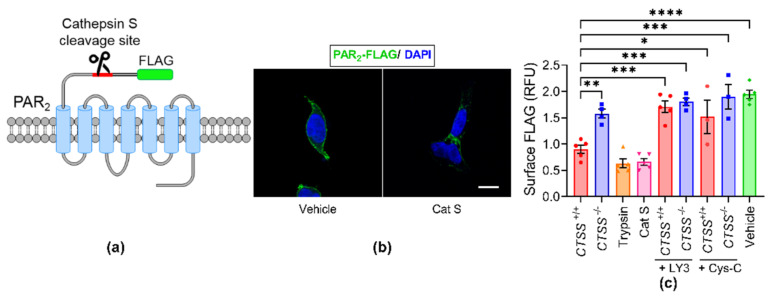
Cathepsin S cleavage of PAR_2_. (**a**) Human PAR_2_ construct showing extracellular FLAG epitope and cathepsin S cleavage site. (**b**) Localization of PAR_2_ using antibodies to extracellular N-terminal FLAG in HEK-FLAG-PAR_2_ cells incubated with vehicle control (vehicle, left image) or cathepsin S (Cat S, right image). Scale bar, 10 µm. (**c**) On-Cell Western assay shows that HSC-3 *CTSS*^+/+^ supernatant, trypsin, and cathepsin S remove the extracellular N-terminal FLAG epitope from HEK-FLAG-PAR_2_ cells. LY3 and cystatin C (Cys-C) inhibit PAR_2_ cleavage in HEK-FLAG-PAR_2_ by HSC-3 *CTSS*^+/+^ and *CTSS*^−/−^ supernatant. One-way ANOVA followed by Tukey’s multiple comparisons test, each data point representing the mean ± S.E.M., * *p* < 0.05, ** *p* < 0.01, *** *p* <0.001, **** *p* < 0.0001 (*n* ≥ 3 cells).

**Figure 8 cancers-13-04697-f008:**
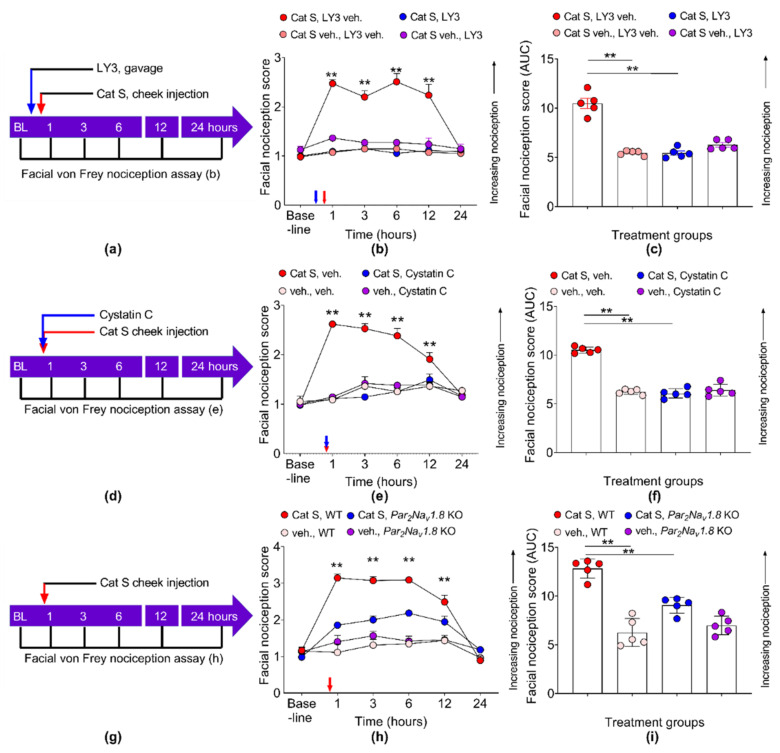
Cathepsin S induces facial nociception through PAR_2_. (**a**,**d**) Experiment timeline for cathepsin S (red arrow) injection into the left cheek of WT C57BL/6J mice or (**g**) Par_2_Na_v_1.8 KO mice. Wild-type mice were treated with (**a**) cathepsin S inhibitor, LY3000328 (LY3) or (**d**) endogenous cathepsin S inhibitor, cystatin C. The facial von Frey nociception assay was conducted before injection (BL or baseline), 1, 3, 6, 12, and 24 h after injection. (**b**) Cathepsin S increased the facial nociception score higher than vehicle injection; the cathepsin S inhibitor LY3 reversed the nociceptive effect of cathepsin S. ** *p* < 0.01, comparison of cathepsin S, cathepsin S inhibitor vehicle to remaining groups at the indicated time point, two-way ANOVA multiple comparisons. *N* = 5 mice in each group. (**c**) Area under the curve (AUC) of each individual mouse in (**b**) was plotted. ** *p* < 0.01, one-way ANOVA. (**e**) Cathepsin S increased the facial nociception score higher than the vehicle; however, cystatin C reversed the nociceptive effect of cathepsin S. ** *p* < 0.01, comparison of cathepsin S, cystatin C vehicle to remaining groups at indicated time point, two-way ANOVA multiple comparisons. *N* = 5 mice in each group. (**f**) AUC of each individual mouse in (**e**) was plotted. ** *p* < 0.01, one-way ANOVA. (**h**) Cathepsin S increased facial nociception score higher than vehicle injection in WT mice, but not in Par_2_Na_v_1.8 KO mice. ** *p* < 0.01 cathepsin S, WT versus cathepsin S, Par_2_Na_v_1.8 KO at indicated time points, two-way ANOVA. *N* = 5 mice in each group. (**i**) The AUC of each individual mouse in (**h**) was plotted. ** *p* < 0.01, one-way ANOVA.

**Figure 9 cancers-13-04697-f009:**
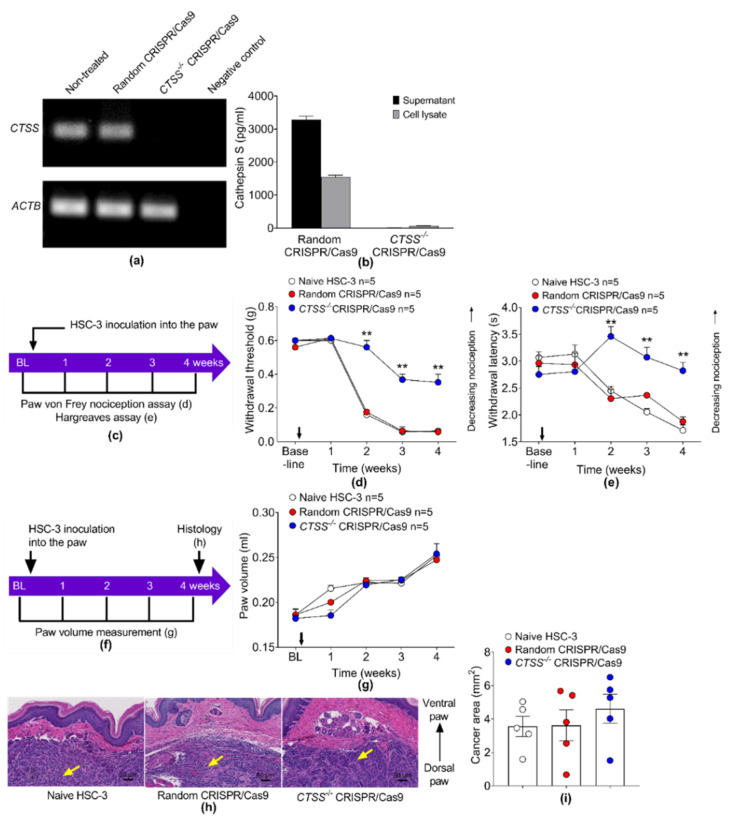
Deletion of *CTSS* with CRISPR/Cas9 in HSC-3 attenuates nociception, but not tumor volume or cancer area, in the mouse oral cancer xenograft model. (**a**) RT-PCR semi-quantitative analysis of *CTSS* expression in non-treated HSC-3 and HSC-3 treated with random CRISPR/Cas9 or *CTSS*^−/−^ CRISPR/Cas9. No *CTSS* mRNA signal was detected in the HSC-3 *CTSS*^−/−^ cells. Beta actin (*ACTB*) was used as a control. (**b**) *CTSS*^−/−^ CRISPR/Cas9 reduced cathepsin S protein in supernatant and whole-cell lysate from HSC-3. (**c**) The experiment timeline for the inoculation of HSC-3 or HSC-3 *CTSS*^−/−^ into the left hind paw of *NU/J Foxn1^nu^* mice. The mice were tested with (**d**) paw von Frey nociception assay and (**e**) Hargreaves thermal assay. Cancer mice generated with HSC-3 *CTSS*^−/−^ exhibited (**d**) a withdrawal threshold and (**e**) a withdrawal latency higher than cancer mice generated with HSC-3 treated with random CRISPR/Cas9. ** *p* < 0.01 HSC-3 *CTSS*^−/−^ mice versus HSC-3 random CRISPR/Cas9 mice, two-way ANOVA. (**f**) In vivo experiment timeline of inoculation of HSC-3 *CTSS*^−/−^ into the paw of *NU/J Foxn1^nu^* mice for the measurement of tumor volume and cancer area. We measured the volume of the cancer paw before inoculation (BL), and every week until 4 weeks after inoculation. The cancer paws were collected for histological H&E staining at 4 weeks after inoculation. (**g**) HSC-3 *CTSS*^−/−^ did not increase cancer paw volume in mice compared to cancer mice inoculated with HSC-3 treated with random CRISPR/Cas9. (**h**) Representative histological H&E images of the cancer (yellow arrow) in the paw inoculated with naive HSC-3 (left image), HSC-3 treated with random CRISPR/Cas9 (middle image), or HSC-3 CTSS^−/−^ (right image). (**i**) Quantification of cancer area in each mouse.

**Figure 10 cancers-13-04697-f010:**
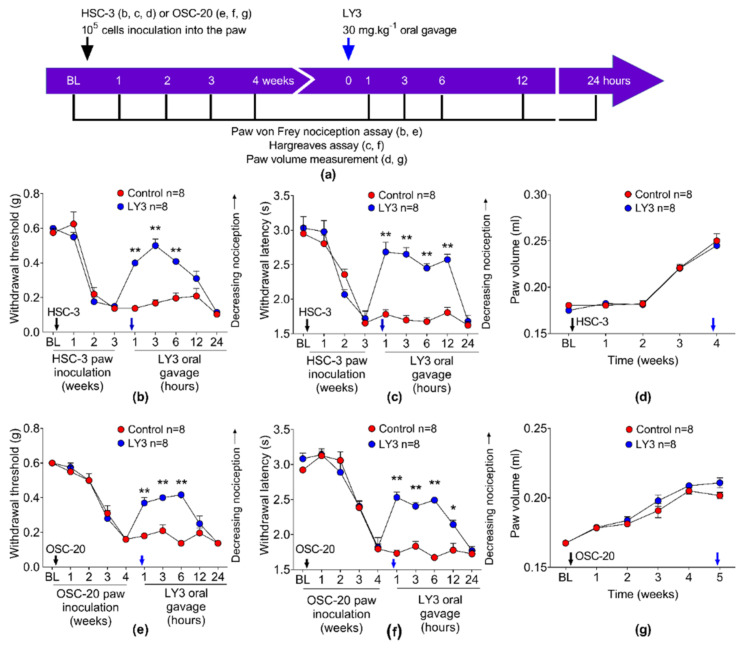
The cathepsin S inhibitor, LY3000328, attenuates nociception in paw xenograft models generated with HSC-3 or OSC-20. (**a**) Experiment timeline for the inoculation of HSC-3 or OSC-20 into the left hind paw of the NU/J *Foxn1nu* mice, subsequent treatment with the cathepsin S inhibitor, LY3000328, and simultaneous nociceptive behavioral testing. The mice were evaluated with (**b**,**e**) the paw von Frey nociception assay, (**c**,**f**) Hargreaves thermal assay, and (**d**,**g**) cancer paw volume before (BL or Baseline) and at 1, 2, 3, and 4 weeks after inoculation to monitor cancer nociception and cancer paw volume. When the mice developed cancer nociception in two consecutive weeks (i.e., week 2, 3 in HSC-3 paw cancer mice or week 3, 4 in OSC-20 paw cancer mice), the mice were administered LY3000328 (blue arrow). The (**b**,**e**) paw von Frey nociception assay and the (**c**,**f**) Hargreaves thermal assay were conducted at 1, 3, 6, 12, and 24 h after LY3000328 treatment. LY3000328 increased (**b**,**e**) withdrawal threshold and (**c**,**f**) withdrawal latency relative to LY3000328 vehicle control (i.e., 30% DMSO in normal saline without LY3000328). (**d**,**g**) LY3000328 did not increase paw cancer volume relative to the control at 24 h after oral gavage of LY3000328. * *p* < 0.05; ** *p* < 0.01 LY3000328 versus control, two-way ANOVA.

**Table 1 cancers-13-04697-t001:** Demographic, anatomic location and tumor staging for patients for which cathepsin S activity was measured in oral SCC and matched normal tissue. Reported pain, as measured by question 7 of the University of California San Francisco Oral Cancer Pain Questionnaire, is provided for each patient.

Patient #	Sex	Age	Ethnicity	Tumor Location	Primary Tumor Stage	Nodal Status	Reported Pain (0–100)
**1**	F	71	Hispanic	Mandibular gingiva	pT4a	pN0	86
**2**	M	57	Hispanic	Mandibular gingiva	pT2	pN2a	92
**3**	M	66	Hispanic	Floor of mouth, Mandibular gingiva	pT4a	pN0	95
**4**	F	77	White/ Non-Hispanic	Mandibular gingiva	pT4a	pN0	86
**5**	F	50	Asian	Tongue	pT1	pN0	10
**6**	M	93	Asian	Mandibular gingiva	pT2	pN0	5
**7**	F	81	White/ Non-Hispanic	Maxillary gingiva	pT2	pN0	74

**Table 2 cancers-13-04697-t002:** Localization of cathepsin S in human oral cancer tissue.

Patient #	Sex	Age	Ethnicity	Tumor Location	Primary Tumor Stage	Nodal Status
A*	F	56	White/Not Hispanic	Tongue	pT4a	pN3b
B	F	75	White/Not Hispanic	Tongue	pT2	pN0
C	M	38	White/Not Hispanic	Tongue	pT2	pN0
D	F	66	White/Not Hispanic	Tongue	pT3	pN1

* The tissue evaluated for patient A includes normal mucosa adjacent to the oral SCC.

## Data Availability

The data are presented in the article.

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
