# Peer review of "Cathepsin S Evokes PAR2-Dependent Pain in Oral Squamous Cell Carcinoma Patients and Preclinical Mouse Models"

_cancers, 2021, doi:10.3390/cancers13184697_

Round 1

Reviewer 1 Report

I suggest to accept.

Reviewer 2 Report

The authors have satisfactorily addressed my concerns. 

This manuscript is a resubmission of an earlier submission. The following is a list of the peer review reports and author responses from that submission.

Round 1

Reviewer 1 Report

This is a very well-written manuscript that convincingly shows a role for Cathepsin S in cancer pain via PAR2.  A variety of approaches were used in both humans and mice, from behavior to molecular biology.  Te experiments are designed appropriately and the data are presented clearly.  The discussion is consistent with the results and appropriate.  This is an important contribution to the field and there are no major problems with the manuscript or the study.  There are a couple of issues that should be clarified.

  1. Since this is a study on cancer pain, and specimens were collected from oral cancer patients, did these patients have pain?  Some information on pain or hyperalgesia in these patients should be provided.
  2. Have there been any adverse effects of Cathepsin S inhibitors in humans?

Reviewer 2 Report

This manuscript entitled “Cathepsin S Evokes PAR2-Dependent Pain in Oral Squamous 2 Cell Carcinoma Patients and Preclinical Mouse Models” shows expression of cathepsin S in oral cancer tissues and a cell line and effects of cathepsin S on hyperexcitability of trigeminal nerves from the mice, and roles of cathepsin S of oral cancer are checked using xenograft models and knockout of CTSS in host or cells. The results are interesting and may contribute to pain management for oral cancer patients. This reviewer has a couple of concerns that should be addressed before recommendation for publication.

(1) In order to show the evidence of the context of the title, expression of PAR2 in oral tissue of xenografts and processing of PAR2 by cathepsin S derived from oral cancer cell lines (in vitro with/without in vivo) should be indicated. Furthermore, PAR2 inhibitor experiments may strengthen the results.

(2) In vitro experiments are performed using only one oral cancer cell line, HSC-3. At least one more cell line should be checked.

Reviewer 3 Report

In this work, Huu Tu, Inoue, and colleagues show that cathepsin S regulates pain in mouse xenograft cancer models, through a PAR2-dependent pathway. This is a timely article in which the authors report that cathepsin S activity and expression are increased in human oral cancer and mouse oral cancer tissue, and that cathepsin S is localized in CD68+ macrophages as well as in keratin-positive cells. By generating xenograft cancer models in tongues from mice, the authors also found that cathepsin S activity and total levels of immunoreactive protein are augmented, but not in a dysplastic oral keratinocyte non-cancer cell line. Because the role of cathepsin S in the trigeminal system had not been previously investigated, in this work they also reported that this lysosomal protease increases trigeminal ganglion neuron excitability, and when PAR2 is absent, excitability is unaltered. Similarly, by selectively deleting PAR2 from Nav1.8-positive neurons, it was shown that cathepsin S-mediated orofacial nociception is dependent on PAR2. Finally, genetic deletion and pharmacologic antagonism of cathepsin S led to the finding that cathepsin S causes nociception in paw xenograft model. The paper is presenting solid experimental results and is well written and structured. Below I have provided a list of questions, comments and minor concerns that arose while reviewing the manuscript:

Minor:

I would suggest changing the title since nociception was only measured in animals. “Cathepsin S Evokes PAR2-Dependent Pain in Preclinical Mouse Xenograft Cancer Models could be a suggestion.

Figures 1 and 4: were the arbitrary units normalized on the expression of the housekeeping protein β-actin? If so, please include the representative blots of β-actin.

Figure 5: In panel “A”, why is the activity of cathepsin B more pronounced in these cell lines than in mouse xenograft model in figure 4? Also, it looks like the activity of cathepsin B and L are also modified in HSC-3. What is the role of these proteases in cancer pain? Please include a brief explanation. In panel “E”, what was the rationale for measuring intracellular and secreted cathepsin S? This outcome is not very well explained or discussed in the manuscript.

Figure 6C: In DRG neurons, cathepsin S is capable of decreasing the rheobase and increasing action potential firing at 2X rheobase. Why in TG, cathepsin S decreasesthe rheobase and does not affect the number of action potentials elicited by twice rheobase current injection, if both parameters measure excitability?

Figure 9G: Since all of the classical ‘hallmarks’ of inflammation have been observed following activation of PARs in vivo, why is the paw volume not altered after deleting cathepsin S? Please explain.

Please include table 3 in the manuscript.

other:

Page 9, line 353: “(CDK5+) should be “CK5+”.

Page 12, line 401: activity was compared beTable 3., please correct the sentence.

Figure 9 is missing subsection “A” in the figure legend, please correct this.

Overall, this study is highly interesting, and the data presented in this study is of significant interest in the field of cancer pain. To conclude, I would accept this paper with some minor modifications.